# Understanding Snail Mucus Biosynthesis and Shell Biomineralisation through Genomic Data Mining of the Reconstructed Carbohydrate and Glycan Metabolic Pathways of the Giant African Snail (*Achatina fulica*)

**DOI:** 10.3390/biology12060836

**Published:** 2023-06-09

**Authors:** Pornpavee Nualnisachol, Pramote Chumnanpuen, Teerasak E-kobon

**Affiliations:** 1Department of Genetics, Faculty of Science, Kasetsart University, Bangkok 10900, Thailand; pornpavee.nu@ku.th; 2Omics Center for Agriculture, Bioresources, Food and Health, Kasetsart University (OmiKU), Bangkok 10900, Thailand; pramote.c@ku.th; 3Department of Zoology, Faculty of Science, Kasetsart University, Bangkok 10900, Thailand

**Keywords:** *Achatina fulica*, land snail, biochemical pathways, glycan, mucus, shell biomineralisation

## Abstract

**Simple Summary:**

In this research, we analysed the genome of the giant African snail to identify enzymes and reconstruct biochemical pathways related to carbohydrate and glycan metabolism. Fourteen complete pathways of carbohydrate metabolism and seven complete pathways of glycan metabolism were found to be associated with the nutrient acquisition and production of the mucus proteoglycans. The enhanced copy number of amylases, cellulases, and chitinases could give the snail an advantage in terms of food consumption and a fast growth rate. The identified ascorbate biosynthesis pathway could be linked to shell biomineralisation. This understanding will benefit the discovery of valuable enzymes for industrial and medical applications.

**Abstract:**

The giant African snail (Order Stylommatophora: Family Achatinidae), *Achatina fulica* (Bowdich, 1822), is the most significant and invasive land snail pest. The ecological adaptability of this snail involves high growth rate, reproductive capacity, and shell and mucus production, driven by several biochemical processes and metabolism. The available genomic information for *A. fulica* provides excellent opportunities to hinder the underlying processes of adaptation, mainly carbohydrate and glycan metabolic pathways toward the shell and mucus formation. The authors analysed the 1.78 Gb draft genomic contigs of *A. fulica* to identify enzyme-coding genes and reconstruct biochemical pathways related to the carbohydrate and glycan metabolism using a designed bioinformatic workflow. Three hundred and seventy-seven enzymes involved in the carbohydrate and glycan metabolic pathways were identified based on the KEGG pathway reference in combination with protein sequence comparison, structural analysis, and manual curation. Fourteen complete pathways of carbohydrate metabolism and seven complete pathways of glycan metabolism supported the nutrient acquisition and production of the mucus proteoglycans. Increased copy numbers of amylases, cellulases, and chitinases highlighted the snail advantage in food consumption and fast growth rate. The ascorbate biosynthesis pathway identified from the carbohydrate metabolic pathways of *A. fulica* was involved in the shell biomineralisation process in association with the collagen protein network, carbonic anhydrases, tyrosinases, and several ion transporters. Thus, our bioinformatic workflow was able to reconstruct carbohydrate metabolism, mucus biosynthesis, and shell biomineralisation pathways from the *A. fulica* genome and transcriptome data. These findings could reveal several evolutionary advantages of the *A. fulica* snail, and will benefit the discovery of valuable enzymes for industrial and medical applications.

## 1. Introduction

*Achatina fulica* (Bowdich, 1822)*,* or the African giant snail, originated in East Africa and is considered an invasive land snail pest of agricultural and ornamental plants [1,2]. Human utilisation through value-added nutraceutical, industrial and medical application is one of the best methods for controlling this snail population and its distribution, with applications including the discovery of novel endoglucanase [3], the use of shell nanoparticles as a greenpoxy biocomposite [4], and the shell CaCO_3_/CaO as the base material of hydroxyapatite [5,6,7]. Gondek et al. [8] also reported that *A. fulica* meat possesses a high protein content (21%) and high levels of unsaturated fatty acids. The mucus of this giant snail has also been widely used in cosmetic supplements and medical biomaterials [9,10,11,12,13,14,15] according to several functional mucous components, such as antimicrobial proteins (83.67 kDa achasin [10], and agglutination-inducing lectin or achatinin [11]), bioactive peptides (11.45 kDa antibacterial mytimacin-AF [16] and anticancer peptides [17,18]), allantoin and glycolic acid [19], and a novel glycosaminoglycan, acharan sulfate, with tumour-suppression activity [20,21]. The snail per se can adjust the mucus composition and properties depending on different activities of the mucous gland cells, i.e., trail mucus for crawling, attachment, communication, protection, and epiphragm mucus during estivation [22,23,24,25]. The snail mucus also plays a role in calcium and mineral biomineralisation during shell formation and repair [26]. Molecular mechanisms and biochemical pathways underlying these intriguing properties of the giant snail remain poorly understood.

Snail genomic research has progressed considerably through the advancement of whole-genome sequencing technologies. These have resulted in an increased number of snail genomes being deposited in the NCBI genome database, i.e., *Biomphalaria glabrata* [27], *Pomacea snails* [28], *A. fulica* [29], *Chrysomallon squamiferum* [30], *Achatina immaculata* [31], *Candidula unifasciata* [32], and *Cepaea nemoralis* [33]. Genetic data analysis with various bioinformatic tools has uncovered several mysteries related to snails and their biological adaptations. For example, long-read and Hi-C data analysis showed more DNA transposons, long terminal repeats, and expansion of particular gene families in the scaly foot snail (*C. squamiferum*), which inhabits deep-sea hydrothermal vents [30]. A comparative genomic analysis of four ampullariid snails (*Lanistes nyassanus*, *P. canaliculata*, *P. maculata*, and *Marisa cornuarietis*) found the expansion of genes involved in cellulolysis and environmental sensing, which could facilitate their aggressively causing damage to agricultural plants [28]. Another recent comparative genomics study of two giant African snail species (*A. fulica* and *A. immaculata*) observed gene duplication events related to terrestrial adaptation, including respiration, estivation, and immune response [31]. The authors also identified the expansion of several gene families, i.e., haemocyanin, phosphoenolpyruvate carboxykinase, zinc metalloproteinase, and mucin-related gene families. Regarding mucus biosynthesis genes, Lie et al. [31] detected 11 heparan sulfate glucosamine 3-O-sulfotransferase (OST) genes essential for the last stage of acharan sulfate biosynthesis and 99 mucin genes encoding core proteins of the mucous proteoglycan. The findings initially suggested some steps and components in the snail mucus biosynthesis pathways.

Our previous research focused on studying the components of giant snail mucus by means of biochemical assays and mass spectrometric analysis, and found different protein and peptide profiles that were unique to the snail species [34]. Bioinformatic prediction of these mucous peptides showed several interesting bioactive activities. These activities were confirmed experimentally, including anti-breast cancer [17] and antibacterial [35] properties. Aside from the protein components of the *A. fulica* mucus, the polysaccharide or glycoconjugate contents of the mucus could be examined by a combination of column chromatography and NMR spectroscopic methods, which were laborious and time-consuming [20]. Cho et al. [36] extracted and characterised bioactive polysaccharides from *A. fulica* using the pressurised hot water method and obtained polysaccharide yields of approximately 42%. Despite these experimental complexities, the biosynthesis of snail mucous polysaccharides and their relevant roles in shell biomineralisation have not been intensively explored. Therefore, this study aimed to identify carbohydrate and glycan metabolic pathways responsible for mucous biosynthesis and shell biomineralisation from the *A. fulica* genomic and transcriptomic data using the KEGG-based biochemical pathway complemented with protein sequence comparison, structural analyses, and manual curation. This analysis could propose potential pathways underlying these biological processes and the snail survival advantages for further experiments and applications.

## 2. Materials and Methods

### 2.1. Preparation of Achatina fulica Genomic Raw Read Data

The genomic DNA of *A. fulica* was previously prepared and sequenced by a paired-end whole-genome Illumina short-read sequencing platform. Sequence read data were quality-checked using FastQC version 12.1 [37]. De novo assembly was performed using MaSuRCA version 4.0.5 [38] and Platanus version 1.3.2 [39]. Genome assembly statistics were computed using QUAST version 4.0 [40]. The *A. fulica* genomic contigs were assessed using BUSCO version 5.4.4 [41] with the metazoa_odb10 dataset to assess the completeness of the assembled genome.

### 2.2. Genome Annotation

Protein-coding genes were predicted from the *A. fulica* genomic contigs and annotated using the Augustus program version 3.4.0 [42]. Functional annotation of the translated protein sequences was performed by searching all predicted sequences against a non-redundant dataset of KEGG GENES [43] and assigning the gene ontology (GO) terms and related KEGG pathways using the GhostKOALA program version 2.2 [44].

### 2.3. Filtration of the Annotated Proteins for Carbohydrate and Glycan-Related Enzymes

Available pathway data (ko00001) from the KEGG database were downloaded by the LINUX command and formatted by regular expression codes written in Python, resulting in a new pathway reference file (KO_Ortholody_ko00001.txt). The functionally annotated *A. fulica* proteins were downloaded from the GhostKOALA program and compared with the KEGG reference pathway data using the Python module named KEGG-to-anvio [45], generating a comparative result file (KeggAnnotations-AnviImportable.txt). Based on the KEGG reference pathways, these annotated proteome data were filtered to find all enzymes related to carbohydrate metabolism and glycan biosynthesis using Microsoft Excel version 16.70. Annotation and classification of these enzymes into different structurally related catalytic and carbohydrate-binding module families based on the CAZy database classification scheme were conducted by dbCAN web server version 5.0 [46].

### 2.4. Reconstruction of the Carbohydrate and Glycan-Related Metabolic Pathways of A. fulica

A list of enzymes from the previous step was employed to reconstruct carbohydrate- and glycan-related metabolic pathways by manually comparing them with the known pathways in the KEGG database. Missing enzymes from the targeted pathways were predicted by BLASTp orthologous search of the reference enzymes against the gastropod and molluscan proteomes to identify those in closely related species that could be matched back with the *A. fulica* proteome using the gene ortholog neighbourhood tool [47], and the resulting orthologs were functionally investigated using the SwissProt/UniProt protein databases. If supporting evidence could not be found for the missing enzymes, a possible hypothesis could be that the reaction was catalysed by an enzyme the protein sequence of which differed from the molluscan ones.

### 2.5. Sequence and Structural Analysis of the Missing Enzyme Candidates

To confirm the BLASTp matches, the protein sequences of the missing enzyme candidates were prepared in the fasta format. The enzyme sequences were used as the input to make the dataset of missing enzyme candidates and blasted against the NCBI protein database using the GALAXY online tools with NCBI BLAST+ makeblastdb and NCBI BLAST+ blastp parameters [47]. The BLASTp results were sorted, and matches with high bit-score and lengths similar to the query sequences were selected. The selected sequence was subjected to protein motif and domain prediction using the InterPro program version 83.0 [48] and protein structure prediction using the SWISS-MODEL program [49], based on the assumption that the structural and domain similarity of the proteins would represent their related biochemical functions. These additional enzymes were then filled into the reconstructed pathways.

### 2.6. Finalising the Reconstructed Metabolic Pathways and Selection of Those Related to Mucous Biosynthesis and Shell Biomineralisation

The mucous biosynthesis and shell biomineralisation pathways were manually checked and selected from the reconstructed carbohydrate-related and glycan metabolic pathways. The results were also compared with the orthologous sequences of *Homo sapiens*, *Lottia gigantea*, *P. canaliculata*, *Crassostrea gigas*, *Mizuhopecten yessoensis*, and *Octopus bimaculoides*. This comparison was used to justify the completeness of the reconstructed pathways and the possibility of undergoing the selected biological processes, i.e., glycosaminoglycan biosynthesis, polysaccharide degradation, and novel biochemical reactions. Manual literature mining was employed to select protein reference sequences for the additional extension of the reconstructed carbohydrate and glycan metabolic pathways.

### 2.7. Preliminary Confirmation of Specific Biochemical Pathways Based on Transcriptomic Analysis of A. fulica Tissues

The Illumina short-read sequencing platforms were used to generate our unpublished transcriptome of the *A. fulica* foot tissues. Sequence read data were quality checked using FastQC version 12.1 [37] and de novo assembled using SPAdes version 3.15.3 [50]. Messenger RNA sequences related to the targeted biochemical pathways were selected using BLASTn search [51] against the NCBI nucleotide database in the GALAXY online tools [52] with NCBI BLAST+ makeblastdb and NCBI BLAST+ blastn parameters. The blast results were sorted and justified based on the high bit-score and the similarity in length. Corresponding protein sequences of the expressed genes were chosen for the protein structure prediction to observe their functional similarity to those with known structures using the SWISS-MODEL program [49].

## 3. Results

### 3.1. Reconstruction of the Carbohydrate and Glycan Metabolic Pathways from the A. fulica Genomic Data

Our gene prediction produced 163,374 protein-coding genes from 312,833 genomic contigs of the *A. fulica* snail, and the GhostKOALA program functionally annotated 22,381 protein sequences (13.7%). Based on the KEGG metabolic pathways, 472 proteins were related to the carbohydrate and glycan metabolic pathways (336 and 136 proteins) (Figure 1). Manual curation was applied to check individual enzymes to remove duplicates and enzymes specific to prokaryotic and plant pathways. Three hundred and sixty-seven enzymes were then used to build the draft carbohydrate and glycan metabolic pathways using the *Homo sapiens* reference pathways. Comparison of the GhostKOALA annotated *A. fulica* proteins to the referenced KEGG pathways confirmed the assignment of 367 enzymes to the draft pathways, and 110 enzymes were selected for the manual curation to discard 62 unrelated proteins (Table 1). Forty-eight proteins were searched against the KEGG database with the organism parameter limited to molluscan species, including *L. gigantea*, *P. canaliculata*, *C. gigas*, *M. yessoensis*, and *O. bimaculoides*, resulting in the addition of 38 enzymes to the *A. fulica* pathways. Reference protein sequences of the remaining ten enzymes were subjected to the sequence similarity search, conserved domain, and structural prediction, allowing the addition of four enzymes for gap filling of the reconstructed pathways. These 377 enzymes were incorporated into 21 complete pathways of the *A. fulica* snail (Figure 2). Six remaining enzymes were missing from three pathways: chondroitin and dermatan sulfate biosynthesis, hyaluronan biosynthesis, and glycosylphosphatidylinositol (GPI) anchor biosynthesis.

The carbohydrate metabolic pathways of the *A. fulica* snail included 14 complete pathways of glycolysis and gluconeogenesis (31 enzymes), citrate cycle (17 enzymes), pentose phosphate pathway (16 enzymes), pentose and glucuronate conversions (nine enzymes), fructose and mannose metabolism (17 enzymes), galactose metabolism (10 enzymes), ascorbate synthesis (nine enzymes), starch and sucrose metabolism (18 enzymes), amino sugar and nucleotide sugar metabolism (23 enzymes), inositol phosphate metabolism (34 enzymes), pyruvate metabolism (18 enzymes), glyoxylate and dicarboxylate metabolism (14 enzymes), propanoate metabolism (15 enzymes), and butanoate metabolism (16 enzymes) as displayed in Figure 2. The glycan-related pathways of the *A. fulica* snail involved seven complete pathways of N-glycan biosynthesis (30 enzymes), O-glycan biosynthesis (seven enzymes), other types of O-glycan biosynthesis (21 enzymes), glycosaminoglycan degradation (13 enzymes), keratan sulfate biosynthesis (nine enzymes), heparin and heparan sulfate biosynthesis (12 enzymes) and N-glycan degradation (nine enzymes). Two incomplete glycan metabolic pathways were involved in chondroitin and dermatan sulfate biosynthesis (11 enzymes), and glycosylphosphatidylinositol (GPI) anchor biosynthesis (18 enzymes), and a member of the hyaluronan biosynthesis pathways was not found. These enzymes and pathways were incorporated into an interaction diagram providing an overview landscape of the carbohydrate and glycan metabolic pathways of the *A. fulica* snail (Figure 2).

### 3.2. Classification of Carbohydrate-Active Enzyme Families from the Reconstructed Carbohydrate and Glycan Metabolic Pathways of A. fulica

From the reconstructed carbohydrate and glycan metabolic pathways, 377 enzymes were classified into 85 families based on the carbohydrate-active enzymes (CAZymes) database (Table 2). Thirty-eight glycoside transferase families were involved in catalysing the transfer of nucleotide diphosphate–sugars, nucleotide monophosphate–sugars, and sugar phosphates and forming the glycosidic bonds. Thirty-one glycoside hydrolase families catalysed the hydrolytic cleavage of the glycosidic bond to produce the carbohydrate hemiacetal, including cellulases, endo-hemicellulases, debranching enzymes, and oligosaccharide-degrading enzymes. Multiple copies of the GH18 (chitinases and endo-β-N-acetylglucosaminidases), GH31 (α-glucosidases, sucrase, isomaltase, α-xylosidases, isomaltosyl-transferases, and maltase/glucoamylases), GH9 (cellulases and endo-glucanases), GH13 (α-glucosidases), and GH1 (β-glucosidases and β-galactosidases) enzyme families were also observed. Six carbohydrate-binding module (CBM) families (CBM13, CBM2, CBM48, CBM14, CBM43, and CBM57) were non-catalytic domains within a long protein sequence, co-occurring with catalytic modules of carbohydrate-active enzymes such as glycoside hydrolases and polysaccharide lyases to enhance their catalytic activities. Four auxiliary activities (AA) families (AA3, AA15, AA2, and AA1) were catalytic proteins potentially involved in cell wall degradation through the assistance of the original glycoside hydrolases (GHs) and polysaccharide lyases (PLs) enzymes. Several members of the esterase families (CE8, CE9, CE10, and CE12) were the enzymes that removed ester-based modifications from carbohydrates. Two polysaccharide lyase families (PL1 and PL4) cleave uronic acid-containing polysaccharides via a β-elimination mechanism to generate an unsaturated hexenuronic acid residue and a new reducing end at the point of cleavage, including pectin lyases and pectate lyases.

### 3.3. Additional Biochemical Pathways That Could Be Important to Mucus Biosynthesis and Shell Biomineralisation in the A. fulica Snail

The analysis of the *A. fulica* reconstructed pathways provided basic biochemical processes for explaining the mucus biosynthesis and shell formation of the *A. fulica* snail. Eight enzymes were identified and the ascorbate biosynthesis pathway essential for shell biomineralisation was completed, i.e., UDP-glucose 6-dehydrogenase (UGDH), glucuronosyltransferase (UGT), myo-inositol oxygenase (MIOX), alcohol dehydrogenase (ADH), gluconolactonase (GNL), gulonolactone oxidase (GULO), dehydroascorbate reductase (DHAR), and ascorbate peroxidase (APX). Our transcriptomic data for *A. fulica* foot tissues confirmed the expression of seven ascorbic acid biosynthesis-related genes, while one enzyme, ascorbate peroxidase, was not detected. Protein structural analysis showed the similarity of these enzymes to those of other molluscan species (Figure 3). Several predicted structures were similar to those of *P. canaliculata*, including 98% coverage and 70.78% similarity for UDP-glucose 6-dehydrogenase, 47% coverage and 38.46% similarity for glucuronosyltransferase, 55% coverage and 34.45% similarity for gluconolactonase, 60% coverage and 53.76% similarity for gulonolactone oxidase, 91% coverage and 71.81% similarity for myo-inositol oxygenase, and 81% coverage and 62.24% similarity for dehydroascorbate reductase. The other two enzymes shared coverage and similarity to other molluscan species, with 93% coverage and 83.55% similarity to alcohol dehydrogenase of *L. gigantea*, and 78% coverage and 49.36% similarity to ascorbate oxidase of *P. caudatus*.

The GhostKOALA annotation result was used to reconstruct the carbohydrate metabolic pathways, and the identification of the ascorbate biosynthesis pathway initiated the linkage to the shell biomineralisation process. Proteins involved in this process were obtained from a literature search on genes related to the shell formation and biomineralisation of molluscs and other invertebrates. Ten groups of proteins were chosen and confirmed in the *A. fulica* proteome using an orthologous sequence search and manual curation (Table 3). Eight of these were mainly ion transporters; increased copy numbers were also present among them, particularly in the case of pore ion channels, Na^+^/sulfate/carboxylate cotransporters, Na^+^/Ca^2+^ exchanger, and Na^+^-dependent ascorbic acid transporter. The hydrolase and oxidoreductase groups also observed high copy numbers of carbonic anhydrases and tyrosinases.

## 4. Discussion

Our bioinformatics analysis workflow was used to analyse the whole genomic contigs of the *A. fulica* snail, and identified enzymes that could be incorporated into this snail’s carbohydrate and glycan metabolic pathways. Using the KEGG pathway references from other animal species, along with the protein sequence and structural analysis, could allow additional enzymes to be manually added, and the drafted pathways to be refined. This suggests the conservation of the core carbohydrate metabolic processes essential for energy and nutrient acquisition (i.e., glycolysis and gluconeogenesis, citrate cycle, pentose phosphate pathway, pentose and glucuronate conversions, fructose and mannose metabolism, galactose metabolism, ascorbate synthesis, starch and sucrose metabolism, amino sugar and nucleotide sugar metabolism, inositol phosphate metabolism, pyruvate metabolism, glyoxylate and dicarboxylate metabolism, propanoate metabolism, and butanoate metabolism). The large copy numbers of certain enzymes highlights the importance of particular biochemical processes in the survival of the *A. fulica* snail, e.g., 89 copies of gluconokinase (GNTK) in the pentose phosphate metabolism, 61 copies of chitinase in the amino and nucleotide metabolism, 28 copies of glucuronosyltransferase (UGT) in the pentose and glucuronate interconversions, and ascorbate biosynthesis, and 25 copies of endoglucanase in the starch and sucrose metabolism.

The molecular activities of the chitinases and endoglucanases allow the *A. fulica* snail to cause drastic damage to agricultural and ornamental plants and chitin-coated organisms (fungi, tiny nematodes, and arthropods) [57]. The detection of 61 copies of the chitinase-coding gene in the *A. fulica* genome suggest the giant snail’s adaptability to omnivorous feeding. Qingliang et al. [57] reported that chitinase was also essential for the molluscan shell biomineralisation and immune system, apart from being the vital enzyme in the visceral mass. CBM14-containing chitinases have been reported to act as antimicrobial proteins to combat chitin-containing pathogens and shown to bind chitin and chitooligomers [58,59]. Cellulose is a linear polysaccharide of glucose residues with β-1, 4-glycosidic linkages and is the structural backbone of the plant cell wall. Cellulose degradation requires multiple enzymes to accommodate this structural heterogeneity. Cellulases (25 copies) of the *A. fulica* snail are divided into three types: (1) endoglucanases for random cleavage of the β-1, 4-glycosidic bonds along the cellulose chain; (2) exoglucanase for cleavage of the nonreducing ends and splitting into elementary fibrils; and (3) β-glucosidase for hydrolysis of cellodextrin and cellobiose to glucose. The CBM2 module identified in this study was reported to facilitate the disruption of crystalline cellulose structure, leading to enhanced cellulase activity [60]. Multiple copies of the AA3-module-containing enzymes in this study have also been reported to be involved in the stimulation of lignocellulose degradation [61]. The *A. fulica* snail synergistically employs these enzymes to quickly digest and complete hydrolysis of cellulose fibres into glucose. Hemicellulose is a heteropolymer composed of diverse sugars categorised into four groups: (1) the pentose sugars (xylose and arabinose); (2) the hexose sugars (glucose, galactose, and mannose); (3) the hexose deoxy sugar (rhamnose); and (4) the acidified sugars (glucuronic acid and galacturonic acid) [62]. Hemicellulose mainly contains pentose sugars, and xylose is the most abundant monosaccharide, and forms the backbone of xylan [63,64]. Hemicellulose degradation requires multiple enzymes to accommodate this structural heterogeneity. Endo-β-1,4-xylanases (EC 3.2.1.8) and β-xylosidases (EC 3.2.1.37) are the enzymes that liberate the side chains from short-branched xylan. The enzymes that cleave polymeric and oligomeric sugars from the xylan backbone are α-l-arabinofuranosidases (EC 3.2.1.55), which remove arabinofuranose residues, β-mannosidase (EC 3.2.1.25), which removes mannose residues, and β-glucosidase (EC 3.2.1.21), which removes 1,4-glucopyranose units [65]. The efficiency of these enzymes would be a critical success factor in giant snail growth. Starch and glycogen degradation is another source of glucose for the energy metabolism of the *A. fulica* snail. Multiple copies of amylases (GH13) and alpha-glucosidases (GH31) help the hydrolysis of the starch at the α-1,4 glycosidic bond into maltose and break down the maltose into glucose. These starch, cellulose, hemicellulose and glycogen degradation pathways of the *A. fulica* snail are summarised in Figure 4. The increase in gene copy numbers related to starch hydrolysis has previously been reported by Axelsson et al. in the context of dog domestication and the adaptation to a starch-rich diet [66]. This gene copy number increment could lead to higher amylase expression, allowing rapid and effective intestinal digestion of high-starch feeds [66]. Glycogen, another substrate for amylases, is the principal means of carbohydrate energy storage in *A. fulica*, and is commonly located in the digestive gland, the foot, and the mantle [67]. Glycogen and galactogen are also found in the secretory cells of the albumen gland [68]. The biosynthesis of glycogen uses UDP-glucose as a common precursor and converts it into activated UDP-glucose using UDP-glucose 4-epimerase. This pathway allows the giant snail to accumulate glycogen during estivation and hibernation [69,70]. Therefore, the glycogen biosynthesis pathway is necessary for the giant snail’s survival under unsuitable environmental conditions.

The glycan biosynthesis and metabolism of *A. fulica* consists of seven complete pathways, including N-glycan biosynthesis, O-glycan biosynthesis, mannose type O-glycan biosynthesis, glycosaminoglycan degradation, keratan sulfate biosynthesis, heparin and heparan sulfate biosynthesis, and N-glycan degradation, as well as three incomplete pathways of chondroitin and dermatan sulfate biosynthesis, hyaluronan biosynthesis, and glycosylphosphatidylinositol (GPI) anchor biosynthesis. These pathways are crucial for the biosynthesis of glycosaminoglycans and proteoglycans, which are the significant components of giant snail mucus. The incomplete chondroitin and dermatan sulfate biosynthesis pathway was found to have two missing enzymes. The first enzyme was beta-1,3-glucuronyltransferase 3 (B3GAT3, EC number 2.4.1.135), which catalyses the transfer of a beta-1,3-linked glucuronic acid to a terminal galactose in precursors of glycosaminoglycan. This study identified an alternative beta-1,3-glucuronyltransferase phosphate (GlcAT-P) with the same EC number as the beta-1,3-glucuronyltransferase 3. The second enzyme was dermatan sulfate 5-epimerase (DSE, EC number 5.1.3.19), which changes the structure at the C-5 terminal of chondroitin-D-glucuronate to dermatan-L-iduronate. An alternative to this enzyme could not be identified in this study, thus suggesting the inability of the *A. fulica* snail to convert chondroitin to dermatan. However, the chondroitin sulfate biosynthesis pathways in *A. fulica* can generate all forms of chondroitin sulfate subunit. Chondroitin sulfate possesses properties related to the immune system and the regulation of inflammation [71]. For this reason, highly sulfated CS, such as CS-C, CS-D, and CS-E units, have anti-inflammatory properties, and CS-A units have both pro- and anti-inflammatory properties [71]. These findings suggest that *A. fulica* GAGs regulate the immune system and are an essential constituent of the snail’s tissues. The hyaluronan biosynthesis pathway had only one missing enzyme, hyaluronan synthase (HAS), which produces hyaluronan from UDP-α-N-acetyl-D-glucosamine and UDP-α-D-glucuronate. This enzymatic reaction is found only in chordates; alternative enzymes could not be identified in this study [72]. The glycosylphosphatidylinositol (GPI) anchor biosynthesis pathway had two missing enzymes. The first enzyme was N-acetylglucosaminyl phosphatidylinositol deacetylase or phosphatidylinositol glycan anchor, class L (PIG-L, EC number 3.5.1.89), which catalyses de-N-acetylation of N-acetylglucosaminyl phosphatidylinositol (GlcNAc-PI) to produce 6-(α-D-glucosaminyl)-1-phosphatidyl-1D-myo-inositol and acetate. The *A. fulica* snail might be able to use N-acetylglucosamine-6-phosphate deacetylase (nagA, EC number 3.5.1.25) instead of PIG-L due to the similar de-N-acetylation catalysis. The second enzyme was GPI mannosyltransferase 1 subunit M, X or phosphatidylinositol glycan anchor, class M, X (PIG-M, X, EC number 2.4.1.-), which catalyses the transfer of the first alpha-1, 4-mannose from dolichol-phosphate-mannose to GlcN-acyl-PI during GPI precursor assembly. Similarly, the *A. fulica* snail might use the GPI mannosyltransferase 2 or phosphatidylinositol glycan anchor, class V (PIG-V, EC number 2.4.1.-) as a replacement due to the same mannosyltransferase activity.

Regarding snail shell biomineralisation, this study identified the ascorbate biosynthesis pathway for vitamin C production from the carbohydrate metabolic pathways of *A. fulica,* as summarised in Figure 5. Vitamin C, which is involved in regulating cholesterol metabolism, acts as a cofactor in various metal ion-dependent enzymes, i.e., collagen and L-carnitine biosynthesis, and potentially the biomineralisation of snail shell [26,73]. There are two important forms of vitamin C, ascorbic acid in a reduced form and dehydroascorbic acid in an oxidised form. In collagen biosynthesis, ascorbic acid plays a key role in the hydroxylation of proline and lysine, which are converted into hydroxyproline and hydroxylysine, which are essential elements of collagen (Figure 6). Multiple types of collagens were also found in the *A. fulica* proteome in this study, including types 1, 6, 9, 12, 14, 15, and 18.

Key enzymes of the ascorbate biosynthesis pathway of the *A. fulica* snail include gulonolactone oxidase (GULO), which catalyses the conversion of L-gulono-lactone to ascorbic acid in the crucial last step and is highly mutated in humans [74], myo-inositol oxygenase, which catalyses the conversion of inositol to glucuronate, and ascorbate oxidase (AO), which converts ascorbate (reduced form) to dehydroascorbate (oxidised form). The vitamin C is absorbed and transported via specific membrane transporters such as solute carrier family 23 proteins found in the *A. fulica* proteome and mediated vitamin C uptake in exchange for sodium ions essential for collagen biosynthesis and perhaps shell calcification (Figure 6). Two vitamin C transporter systems have been identified, i.e., the glucose transporter (GLUT or SLC2A) specific to dehydroascorbic acid and the sodium-dependent vitamin C transporter (SVCT) specific to ascorbic acid. Two isoforms of the SVCT or Na^+^-coupled cotransporter proteins were also found in this study, including SVCT1 (SLC23A1) and SVCT2 (SLC23A2) [75]. The ascorbic acid is transported into the rough endoplasmic reticulum and facilitates the hydroxylation of the procollagen precursor, which is then glycosylated by galactosyltransferase and glucosyltransferase to form tropocollagen. The tropocollagen is later exocytosed to the extracellular side for further modification and crosslinking to form the extracellular matrix. This fibrous collagen network initiates a scaffold for biomineralisation.

As an invertebrate with an open circulatory system, the giant snail uses body fluid or haemolymph to circulate substances between cells throughout the body or accumulate in the haemocoel, a compartment inserted into tissues and organs. The haemolymph also contains water, O_2_, CO_2_, waste products, inorganic salts (Na^+^, K^+^, Ca^2+^, Cl^−^, HCO_3_^−^), organic compounds (carbohydrates, proteins, and lipids), hormones, haemocyanin, and free suspended cells called haemocytes [76,77]. Rousseau (2003) reported the presence of three main calcium-enriched compartments in the black-lip pearl oyster (*Pinctada margaritifera),* the gill, the haemolymph and the mantle, that are involved in the shell biomineralisation process [78]. The haemolymph was considered to facilitate the passage of high concentrations of Ca^2+^ and HCO_3_^−^, and served as the calcium pool.

Mollusc shells contain 95% calcium carbonate (CaCO_3_) and 5% organic matrix [79]. The molluscan shell formation primarily involves mineral transport from the mantle tissue to the mucus-mediated extracellular space. Then, the mineral components are laid down as organised crystals onto the collagen-containing organic matrix [60]. In this study, several ion transporter proteins were identified that could be relevant to the biomineralisation of the *A. fulica* shell, using previously published research as references. Molluscans are known to have three ion transporter proteins (SLC23A1, SLC8A, and SLC26A) that are involved in biomineralisation, and these proteins also appear in the *A. fulica* proteome [55]. First, SLC23A1, or solute carrier family 23 member 1, transports the ascorbic acid in exchange for sodium ions. Rosenthal et al. [80] reported that SLC23A1 was upregulated during larval calcification in corals, suggesting that similar activity could occur during snail shell calcification [80]. Second, SLC8A or sodium-calcium exchanger 3 is the primary exchanger extruding Ca^2+^ from the cell, and might be involved in the shell biomineralisation [81]. Third, SLC26A, or sodium-independent transporter, is an anion channel that transports various ions, including sulfate (SO_4_^2−^), oxalate ((C_2_O_4_)^2−^), iodine (I^−^), chloride (Cl^−^), formate (HCO_2_^−^), and bicarbonate (HCO_3_^−^). The bicarbonate ions are required for the formation of calcium carbonate crystals. Capasso et al. [53] found that the bicarbonate transporter SLC4 regulates intracellular bicarbonate ion homeostasis to buffer excess H^+^ generated during CaCO_3_ precipitation and supply of HCO_3_^−^ to the calcifying cells of the mantle tissues [53,82].

The *A. fulica* shell is typically composed of three essential layers: an external periostracum layer, a thick middle layer called the prismatic layer with a steep bulge of calcium carbonate crystals tightly packed together, and an inner nacreous layer. Shitalbahen (2004) reported observing collagen fibres coated on both the nacreous and prismatic layers of the Eastern oyster *Crassostrea virginica* [83]. This evidence highlights the importance of the previously discussed ascorbic biosynthesis pathway in collagen biosynthesis, which could later be exported and become a scaffold for calcium carbonate crystallisation (Figure 7). The extrapallial space (EPS) is a space between the mantle and the shell, and is filled with supersaturated extrapallial fluid (EPF) containing precursor ions of the calcium carbonate, as well as other ions such as Na^+^, K^+^, Mg^2+^, Cl^−^, and SO_4_^2−^ [84]. The giant snail ingests the calcium from food before the calcium is absorbed into the haemolymph. The calcium ions are secreted into the extrapallial or extracellular space passively by diffusion through the Ca^2+^ transporter or active passage by the Ca^2+^ pumps. In contrast, bicarbonate ions are actively transported to the EPS through SLC26A transporters or HCO_3_^−^/Cl^−^ exchangers and SLC4A transporters or HCO_3_^−^/Na^+^ exchangers. The source of bicarbonate ions is the carbon dioxide released from the energy metabolic pathways. Carbonic anhydrase catalyses the hydration of carbon dioxide into bicarbonate (the source of inorganic carbon for CaCO_3_ precipitation) and hydrogen ions, which must be removed from the EFS to prevent the calcium carbonate dissolving, suggesting a direct role in shell calcification [80,85]. This study found 30 copies of the carbonic anhydrase coding genes. These enzymes would catalyse the fast-growing shell of the *A. fulica* snail by combining the bicarbonate and calcium ions accumulated in the EPS. For the outermost periostracum layer, tyrosinase catalyses the production of quinones attached to amino acid, forming insoluble proteins of this uncalcified proteinaceous cuticle layer to seal EPS from the external environment [86]. Another Na^+^-coupled transporter SLC13A has also been identified, and possesses a widely known calcification role in vertebrates [87]. Therefore, the shell calcification pathways of the *A. fulica* snail are proposed to be as shown in Figure 7, providing a basic understanding for the further investigation of shell shape and pattern formation in terrestrial molluscs.

## 5. Conclusions

This study successfully reconstructed fourteen carbohydrate and seven glycan metabolic pathways from the draft genomic contigs of the *A. fulica* snail using bioinformatic analysis and manual curation. Alternative enzymes were suggested for two incomplete pathways. Two enzymes were missed from the glycan metabolic pathways, suggesting the possible inability of hyaluronan and dermatan biosynthesis compared to those of the vertebrate reference. Mining these carbohydrate and glycan metabolic pathways could explain the survival capability of the *A. fulica* snail. The increased copy numbers of amylases, cellulases, chitinases, and other glycoside hydrolases in the carbohydrate metabolic pathways allowed the snail to grow rapidly and consume a wide range of foods. Identification of the ascorbate biosynthesis pathway could be extended by literature mining to propose the snail shell biomineralisation process associated with collagen networks, carbonic anhydrases, tyrosinases, and several ion transporters. Mining the glycan metabolic pathways could preliminarily determine the basic glycosaminoglycans produced within the *A. fulica* mucus (keratan sulfate, heparan sulfate, acharan sulfate, and chondroitin sulfate). Therefore, this genomic data mining could explain several vital evolutionary advantages of the *A. fulica* snail. The enzymes and pathways identified in this study should be validated and tested for further medical and industrial applications.

## Figures and Tables

**Figure 1 biology-12-00836-f001:**
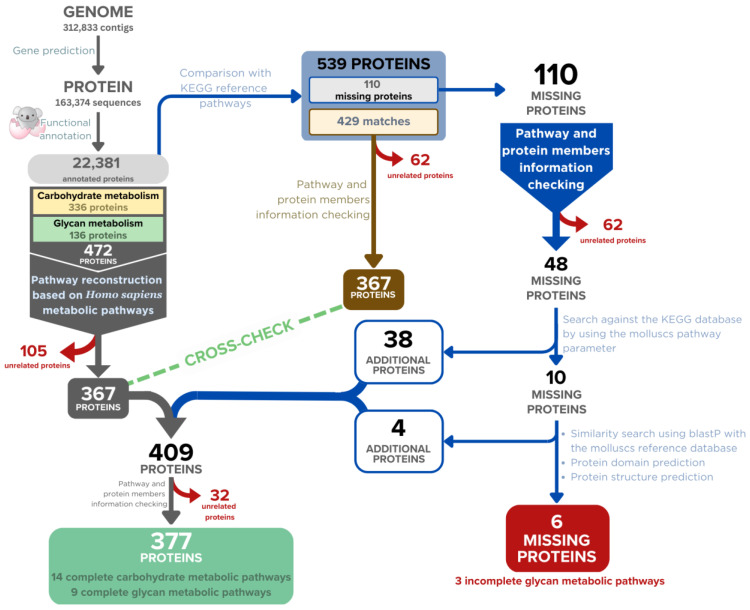
Overview of the bioinformatic workflow for reconstructing the carbohydrate and glycan metabolic pathways of the *A. fulica* snail based on the GhostKOALA assignment (grey arrows). The 367 proteins were confirmed by cross-checking with pathway and protein member information (brown arrow). The pathway gaps are successfully filled with additional enzymes by adjusting the KEGG pathway search parameters, manual curation, and protein domain and structure prediction (blue arrows). The unrelated proteins were removed from the candidate list (red arrows).

**Figure 2 biology-12-00836-f002:**
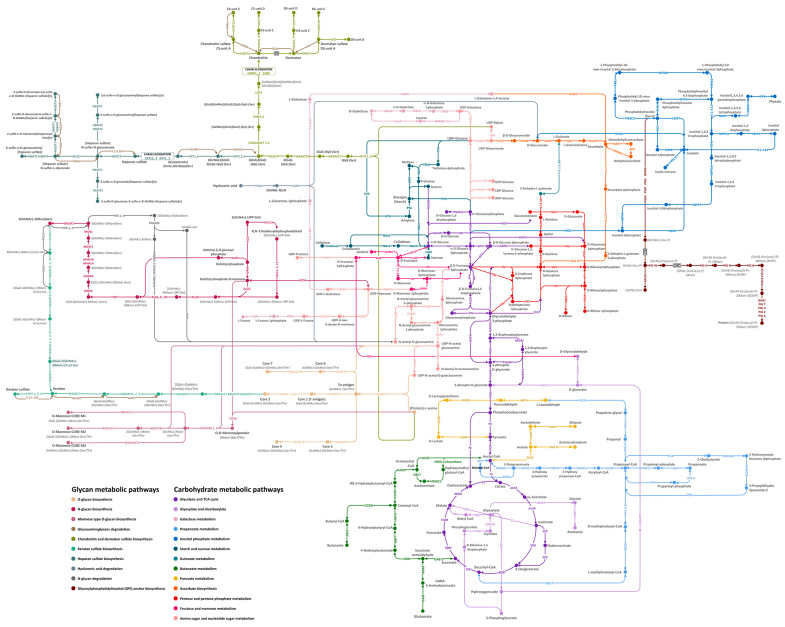
Overview of the carbohydrate and glycan metabolic pathways of the *Achatina fulica* snail reconstructed by the genomic contig data and functionally annotated based on the KEGG pathway database followed by three-step manual curation to fill the additional enzymes into the missing gaps. The interaction graph indicates the enzymes with solid dots, the enzymatic direction with arrows, pathway annotation with different colours, intermediates or metabolites with black characters, and enzyme names with coloured abbreviated names. Text labels show the pathway names.

**Figure 3 biology-12-00836-f003:**
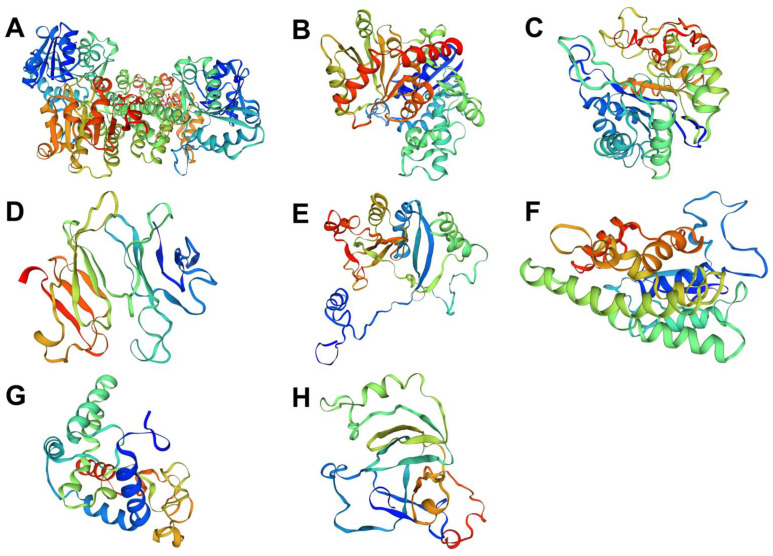
Predicted three-dimensional structures of enzymes related to the ascorbate biosynthesis pathway of *A. fulica* using the homology prediction method and templates from related molluscan species. These enzymes consist of (**A**) UDP-glucose 6-dehydrogenase (UGDH), (**B**) glucuronosyltransferase (UGT), (**C**) alcohol dehydrogenase (ADH), (**D**) gluconolactonase (GNL), (**E**) gulonolactone oxidase (GULO), (**F**) dehydroascorbate reductase (DHAR), (**G**) myo-inositol oxygenase (MIOX), and (**H**) ascorbate oxidase (AO).

**Figure 4 biology-12-00836-f004:**
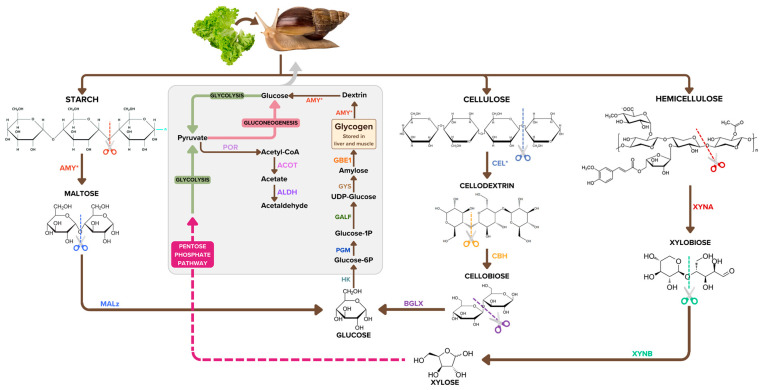
Starch, cellulose, hemicellulose, and glycogen degradation pathways of *Achatina fulica* were reconstructed from our bioinformatic analysis to perform metabolic pathway prediction on the basis of the giant snail genome. The pathways include enzyme names in abbreviation, metabolite intermediates with some structural representation, and arrows determining the reaction direction. Polysaccharide degradation, glycolysis, gluconeogenesis, and pentose phosphate pathways were represented by brown, green, pale pink, and bright pink arrows. Asterisks (*) indicate the enzymes with high copy numbers.

**Figure 5 biology-12-00836-f005:**
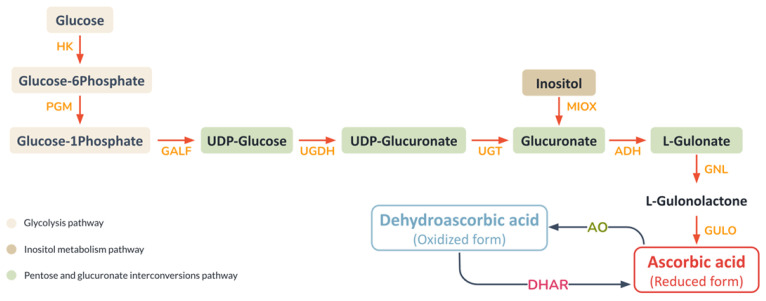
Proposed ascorbic acid biosynthesis pathway of *A. fulica* extracted from the reconstructed carbohydrate pathways. This pathway includes intermediate and abbreviated enzyme names, arrows for the reaction direction, and relevant biochemical pathways in different coloured boxes.

**Figure 6 biology-12-00836-f006:**
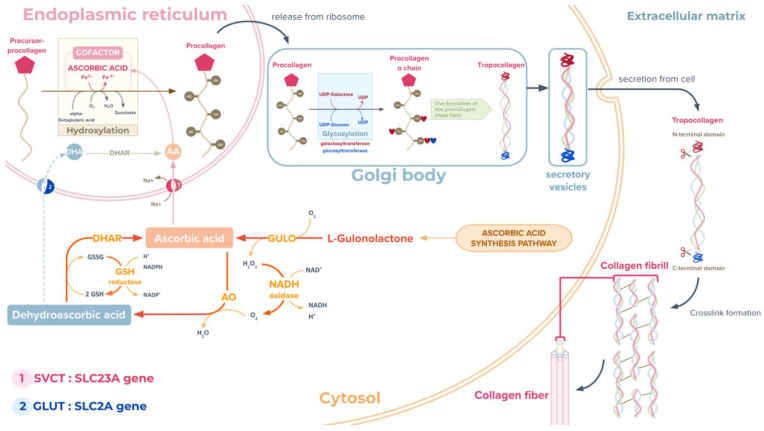
Proposed ascorbic acid biosynthesis pathway of the *Achatina fulica* snail with collagen protein biosynthesis and modification. The synthesised ascorbic acid is transported into the endoplasmic reticulum and facilitates the hydroxylation of the procollagen precursor before being glycosylated to form tropocollagen and later released to the extracellular side for further modification and crosslinking (grey arrows). The ascorbic acid transport is driven by sodium-dependent vitamin C transporters or SVCTs (pink circles and pink arrows) and glucose transporters or GLUTs (blue circles and a blue arrow) for dehydroascorbic acid. DHAR and AO control interconversion between the dehydroascorbic acid and ascorbic acid (orange arrows).

**Figure 7 biology-12-00836-f007:**
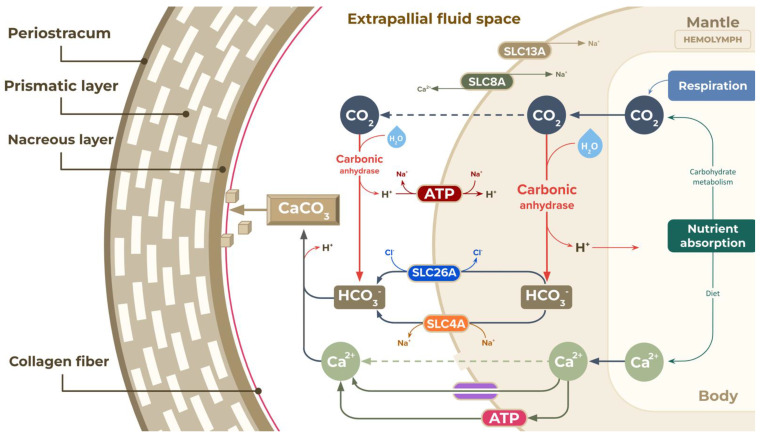
Proposed pathways of the shell calcification and biomineralisation in *Achatina fulica*. The primary source was the release of CO_2_ from the carbohydrate metabolism (green arrow) and transport to the extrapallial fluid space (EFS) (grey dashed arrow). The CO_2_ was converted to HCO_3_^−^ by several intracellular carbonic anhydrases (red arrow), and the bicarbonate ions were actively transported to the EFS through SLC26A transporters or HCO_3_^−^/Cl^−^ exchangers (dark blue oval and dark brown arrows) and SLC4A transporters or HCO_3_^−^/Na^+^ exchangers (orange oval and dark brown arrow). Excess hydrogen ions produced by the formation of calcium carbonate and from carbonic anhydrase were eliminated through the Na^+^/H^+^ exchanger (dark red oval and dark red arrow) out of the EFS. Calcium intake from food could reach the extrapallial fluid space passively by diffusion through Ca^2+^ channels (purple oval and dark green arrow) and gap junctions (green dash arrow) or actively by the Ca^2+^-pumping ATPase (pink oval and dark green arrow). The CaCO_3_ then formed the structure of prismatic and nacreous layers making up the *A. fulica* shell with the assistance of a collagen protein network as the scaffold (red line).

**Table 1 biology-12-00836-t001:** The number of enzymes missed from the reconstructed carbohydrate and glycan metabolic pathways of the *A. fulica* snail. Additional enzymes were successfully filled into the pathways after three curation steps (manual curation, sequence similarity search, and conserved domain and structural prediction). Numbers represent the number of enzymes, and ✓ indicates that all enzymes are present in that particular pathway.

Pathways	Number of Missing Enzymes	Number of Enzymes That Passed the Filtering Criteria	Number of Enzymes Added After the First Gap Filling	Number of Enzymes Added After the Second Gap Filling	Completed Pathways/Remained Missing Enzymes
Carbohydrate metabolic pathways					
Pentose phosphate	12	8	4	-	✓
Pentose and glucuronate interconversions	9	8	-	1	✓
Fructose and mannose metabolism	7	4	3	-	✓
Galactose metabolism	3	2	1	-	✓
Ascorbate metabolism	5	4	1	-	✓
Starch and sucrose metabolism	5	3	2	-	✓
Amino sugar and nucleotide sugar metabolism	3	3	-	-	✓
Inositol phosphate metabolism	8	5	3	-	✓
Glyoxylate and dicarboxylate metabolism	11	8	3	-	✓
Propanoate metabolism	6	3	3	-	✓
Butanoate metabolism	6	3	3	-	✓
Pyruvate metabolism	5	4	-	1	✓
Glycolysis/gluconeogenesis	4	1	3	-	✓
Glycan metabolic pathways					
Chondroitin sulfate/dermatan sulfate biosynthesis	4	-	2	-	2
Galactose type of O-glycan biosynthesis	1	-	1	-	✓
Glycosylphosphatidylinositol (GPI) biosynthesis	4	-	1	-	3
Mannose type O-glycan biosynthesis CORE 1	5	4	1	-	✓
Mannose type O-glycan biosynthesis CORE 2	3	1	2	-	✓
Mucin type of O-glycan biosynthesis	2	1	1	-	✓
N-glycan biosynthesis	1	-	1	-	✓
Glycohormone type of N-glycan biosynthesis	3	-	2	-	✓
Hyaluronan biosynthesis	1	-	-	-	1
Keratan sulfate biosynthesis	2	-	-	2	✓
Total number of enzymes	110	48	38	10	6

**Table 2 biology-12-00836-t002:** The *A. fulica* enzymes (377 genes) were assigned from the reconstructed carbohydrate and glycan metabolic pathways based on the carbohydrate-active enzymes (CAZymes) database using dbCAN version 5.0. The number of the protein subfamilies and the associated numbers of each subfamily are presented. The protein subfamily names are displayed as abbreviations.

Protein Family Names	Number of Protein Subfamilies	Protein Subfamily Names
Auxiliary activities (AAs)	4	AA1 (1), AA2 (1), AA3 (7), AA15(8)
Carbohydrate-binding modules (CBMs)	6	CBM2 (3), CBM13 (16), CBM14 (2), CBM43 (1), CBM48 (2), CBM57 (1)
Carbohydrate esterases (CEs)	4	CE8 (1), CE9 (1), CE10 (5), CE12 (1)
Glycoside hydrolases (GHs)	31	GH0 (1), GH1 (6),GH2 (8), GH7 (1), GH9 (7), GH13 (8), GH15 (1), GH18 (12), GH19 (1), GH20 (5) GH23 (1), GH25 (1), GH27 (2), GH29 (3), GH31 (10), GH35 (3), GH36 (1), GH37 (2), GH38 (8), GH39 (1), GH47 (4), GH48 (1), GH56 (1), GH57 (1), GH59 (1), GH63 (2), GH79 (1), GH84 (1), GH85 (2), GH89 (1), GH99(1)
Glycosyltransferases (GTs)	38	GT1 (10), GT2 (19), GT3 (1), GT4 (7), GT7 (14), GT8 (2), GT10 (3), GT11 (11), GT13 (2), GT14 (7), GT15 (1), GT16 (1), GT18 (1), GT20 (1), GT22 (4), GT23 (4), GT25 (1), GT27 (19), GT29 (1), GT30 (1), GT31 (22), GT33 (1), GT35 (3), GT39 (1), GT41 (4), GT47 (2), GT49 (1), GT54 (2), GT57 (2), GT59 (1), GT61 (1), GT64 (3), GT65 (1), GT66 (3), GT68 (1), GT92 (4), GT95 (2), GT98 (1)
Polysaccharide lyases (PLs)	2	PL1 (1), PL14 (1)

**Table 3 biology-12-00836-t003:** List of proteins related to shell biomineralisation and calcification of the *A. fulica* snail. Copy numbers are indicated in brackets, and superscript letters represent previous research studies on the biomineralisation of molluscs and other invertebrates.

Protein Groups	Protein Names (Copy Number of Proteins)
Bicarbonate transporter ^a^	Solute carrier family 4 (anion exchanger), member 2 (5), Solute carrier family 4 (sodium bicarbonate cotransporter), member 7 (2), Solute carrier family 4 (sodium bicarbonate cotransporter), member 8 (5), Solute carrier family 4 (sodium bicarbonate cotransporter), member 10 (3), Solute carrier family 4 (sodium bicarbonate cotransporter), member 11 (2)
Pores ion channels ^a^	Ammonium transporter (32), Glutathione S-transferase (23)
Na^+^/sulfate/carboxylate cotransporter ^a^	Solute carrier family 13 (sodium-dependent dicarboxylate transporter), member 2/3/5 (10)
Facilitative GLUT transporter ^b^	Solute carrier family 2 (facilitated glucose transporter), member 1 (3), Solute carrier family 2 (facilitated glucose transporter), member 8 (1), Solute carrier family 2 (facilitated glucose transporter), member 11 (1)
Na^+^/Ca^2+^ exchanger ^a^	Solute carrier family 8 (sodium/calcium exchanger) (15)
Multifunctional anion exchanger ^a^	Solute carrier family 26 (sulfate anion transporter), member 1 (1), Solute carrier family 26 (sulfate anion transporter), member 2 (1), Solute carrier family 26 (sulfate anion transporter), member 5 (2), Solute carrier family 26 (sulfate anion transporter), member 6 (4), Solute carrier family 26 (sulfate anion transporter), member 7 (2), Solute carrier family 26 (sulfate anion transporter), member 10 (1), Solute carrier family 26 (sulfate anion transporter), member 11 (2)
Hydrolyases ^c^	Carbonic anhydrase (4.2.1.1) (30)
Ca^2+^ transporter ^d^	P-type Ca^2+^ transporter type 2A (5), P-type Ca^2+^ transporter type 2B (3), P-type Ca^2+^ transporter type 2C (2)
Oxidoreductases ^c^	Tyrosinase (16)
Na^+^-dependent ascorbic acid transporter ^a^	Solute carrier family 23 (nucleobase transporter), member 1 (10)

^a^: Capasso et al. [53], ^b^: Przybyło and Langner [54], ^c^: Sleight et al. [55], and ^d^: Louis et al. [56].

## Data Availability

The relevant genomic and transcriptomic data are available via this link (https://tinyurl.com/5n6jemzj).

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
