# Peer review of "Understanding Snail Mucus Biosynthesis and Shell Biomineralisation through Genomic Data Mining of the Reconstructed Carbohydrate and Glycan Metabolic Pathways of the Giant African Snail (Achatina fulica)"

_biology, 2023, doi:10.3390/biology12060836_

Round 1

Reviewer 1 Report

Please see the marked pdf file. 

Author Response

Thank you very much for your comments. The authors edited and revised the manuscript according to your comments (as highlighted in the attached file). Please see the attachment.

Reviewer 2 Report

The manuscript biology-2393662 investigated the snail mucus biosynthesis and shell biomineralisation for example of popular land mollusk Achatina fulica using modern bioinformatic workflow.  Received genomic data mining could explain several vital evolutionary advantages of the snails. This study’s identified enzymes and pathways could be validated and tested for further medical and industrial applications. The paper is well written, organized and adds new understanding to the biology of the most significant and invasive land snail pest. I do not have any major concerns, and recommend consideration for inclusion in BIOLOGY after elimination of small remarks.

1.                 The manuscript talks a lot about the metabolic pathways of such common carbohydrates as cellulose, chitin and starch. At the same time, there are other important natural polysaccharides, such as hemicelluloses, in particular, xylan can be found in plants up to 30% by weight. It is advisable to strengthen the information about this with a few phrases and give the study a greater globality. One of the references to this in the manuscript is Line 231, misspelled, correct: endo-hemicellulAses. Please correct.

2.                 Certainly an important part of the manuscript - Figure 2 is absolutely unreadable. It is proposed to improve the quality, or break it into fragments and transfer it to Supplementary materials.

Author Response

The authors would like to thank you for the comments from the reviewer. We addressed your comments carefully, as highlighted in the attached file.

  1. The manuscript talks a lot about the metabolic pathways of such common carbohydrates as cellulose, chitin and starch. At the same time, there are other important natural polysaccharides, such as hemicelluloses, in particular, xylan can be found in plants up to 30% by weight. It is advisable to strengthen the information about this with a few phrases and give the study a greater globality. One of the references to this in the manuscript is Line 231, misspelled, correct: endo-hemicellulAses. Please correct. Answer: The authors added more discussion on the hemicelluloses and their related degrading enzymes. Figure 4 was also modified to cover this part corresponding to our results.
  2. Certainly an important part of the manuscript - Figure 2 is absolutely unreadable. It is proposed to improve the quality, or break it into fragments and transfer it to Supplementary materials. Answer: The in-manuscript figure 2 might give an overview landscape of the pathways. The higher resolution file is provided where the readers can expand and read all details in the figure (https://tinyurl.com/2scrwjxk). 
